# Erosion–Corrosion of Novel Electroless Ni-P-NiTi Composite Coating

Rielle Jensen [1,*], Zoheir Farhat [1], Md. Aminul Islam [2] and George Jarjoura [1]

1   Department of Mechanical Engineering, Dalhousie University, 1360 Barrington Street, Halifax, NS B3J 2X4, Canada
2   National Research Council Canada, 4250 Wesbrook Mall, Vancouver, BC V6T 1W5, Canada
*   Correspondence: rielle.jensen@dal.ca

**Abstract:** The lifespan of low-carbon steel petroleum pipelines can often be shortened by the erosion–corrosion damage caused by their service conditions. Applying electroless Ni-P coating is a promising option to protect the steel from the environment due to its high hardness and corrosion resistance. However, electroless Ni-P has a low toughness but can be increased by the addition of NiTi ductile particles. This work produced electroless Ni-P and Ni-P-NiTi coatings of different thicknesses on AISI 1018 substrates and compared their erosion, corrosion, and erosion–corrosion behaviors. The methodology involved conducting slurry pot erosion–corrosion tests on AISI 1018 steel substrate, the monolithic Ni-P coatings, and the composite Ni-P-NiTi coatings. Erosion resistance was highly influenced by coating thickness, presumably because of the relationship between the erosion-induced compressive stresses and the coating's as-plated internal stresses. The NiTi nanoparticle addition was highly effective at improving the erosion–corrosion resistance of the coating. Pitting corrosion and cracking were present after erosion–corrosion on the monolithic Ni-P coatings. However, the Ni-P-NiTi composite coating had a relatively uniform material loss. Overall, the AISI 1018 steel substrate had the worst erosion–corrosion resistance and 25 μm thick Ni-P-NiTi coating had the best.

**Keywords:** Ni-P-NiTi composite coating; electroless nickel phosphorous; coating thickness; erosion–corrosion; synergy

## 1. Introduction

Pipelines are the most convenient way for the transportation of large volumes of oil and gas over long distances to meet the high demand for petroleum products [1,2]. Low-carbon steel is a commonly used pipeline material due to its strength, durability, and wide availability [3]. Corrosive species, such as $Cl^-$, $O_2$, $H_2S$, or $CO_2$, are often present in petroleum. Additionally, particulates, for instance, sand or other solid particles, could also be present. As a result, the pipeline can significantly degrade the steel through corrosion and erosion [2,4,5]. When the mechanical abrasion and electrochemical corrosion are coupled, a material loss mechanism known as erosion–corrosion occurs [2,6]. This results in a synergistic effect, where there is a higher mass loss than the summation of the mass loss from pure erosion and pure corrosion acting separately [1,6,7]. Therefore, the material degrades at a faster rate which reduces the steel pipeline's lifespan [8]. The frequent replacement of damaged pipeline steel is not efficient, and wear prevention through high durability material alternatives is costly [2,8]. A promising method for slowing the rate of metal loss is the use of surface enhancement to protect the base material from the environment [5,9–13]. A seemingly suitable option is an electroless nickel-prosperous (Ni-P) coating because of its exceptional adhesion and high corrosion resistance from the lack of grain boundaries [2,5,10,14]. This coating has already been widely used as a protective coating in many industries. However, Ni-P has a low toughness which would make it particularly susceptible to wear from erosion. Increasing the toughness while

maintaining Ni-Ps high adhesion and corrosion resistance can be achieved through the addition of ductile particles [2,14,15].

When ductile particles are added into the matrix of a brittle material such as Ni-P coating, the increase in toughness can be explained through the crack's interaction with the particles [16,17]. There are different types of particle-crack interactions; however, all of the interactions absorb some of the crack's propagation energy, causing the crack to lose its driving force. These toughening mechanisms include crack bridging, crack deflection, micro-cracking, and transformation toughening [18,19].

Transformation toughening occurs for particles that have stress-induced phase transformation mechanisms, such as the nickel–titanium (NiTi) alloy [16,20]. When the crack interacts with the particle, it applies stress that causes a phase transformation. The transformation itself absorbs crack propagation energy. Additionally, the increase in the volume of the particle during the transformation creates a compressive stress field in the matrix surrounding the particle. This stress field reduces the tensile stresses involved in the crack opening [19,20].

The addition of ductile NiTi nanoparticles has been shown as a promising candidate for improving the toughness of the Ni-P coating for the use of oil and gas pipelines [16]. Li et al. [14] studied the erosion–corrosion resistance of an electroless Ni-P-Ti coating, and the ductile titanium (Ti) particle's addition significantly improved the pure corrosion, pure erosion, and erosion–corrosion resistance. In another study, after the Ni-P-Ti coating was produced, Li et al. [21] used heat treatment to form NiTi through diffusion and studied its erosion–corrosion behavior. This study showed promise for NiTi particles to improve the erosion–corrosion resistance of an electroless Ni-P coating, but there are unknowns on how it is affected by different coating properties. First, since the formation of NiTi particles involved the heat treatment of Ni-P-Ti, it is unclear how the electroless Ni-P coating as-plated with NiTi nanoparticles performs. Furthermore, the erosion–corrosion resistance of the composite coating as a function of the coating thickness has not been investigated. When the authors Jensen et al. [22] previously investigated the wear behavior of the as-plated electroless Ni-P-NiTi composite coating, it was found that the wear resistance of the coating was greatly influenced by its thickness. It is therefore likely that the coating thickness also influences its erosion–corrosion behavior.

In general, coatings have high internal tensile stress near the substrate. As the thickness increases, the tensile stress decreases rapidly and at a higher thickness, compressive stress develops [23]. For instance, Saraloğlu Güler et al. [24] studied how residual stress was affected by the electrodeposition parameters of electrodeposited nickel (Ni) and Ni–$MoS_2$ composite coatings. The findings included that increasing the thickness and molybdenum-disulfide ($MoS_2$) addition both resulted in a decrease in the internal tensile stress values. The particle additions also contributed to a change in the nature of the internal stresses from tensile to compressive [24].

The objective of this work is to study the erosion–corrosion behavior of electroless Ni-P and Ni-P-NiTi coatings of various thicknesses. The effect that the addition of NiTi nanoparticles to an electroless Ni-P coating has on its erosion–corrosion resistance is examined, as well as the extent that the thickness affects the erosion–corrosion behaviors of the Ni-P and Ni-P composite coatings.

## 2. Materials and Methods

### 2.1. Coating Preparation

Rectangular AISI 1018 steel coupons were used as a coating substrate and as a reference substrate. The chemical composition of the steel is shown below in Table 1 [25]. The coupons had the dimensions of 18 mm × 10 mm × 6 mm and were thoroughly polished before coating.

**Table 1.** Chemical Composition of AISI 1018 Steel [25].

| Weight % | AISI 1018 |
|----------|-----------|
| C | 0.182 |
| Mn | 0.754 |
| Cu | 0.186 |
| Cr | 0.181 |
| Si | 0.095 |
| P | 0.040 |
| Fe | Balance |

All steel specimens were ground using 240, 320, 400, and 600 grit SiC papers. The ground substrates were then polished using diamond solutions of 9 μm, 3 μm, and 1 μm, respectively.

Industrial-grade Ni-P solution was used as a plating solution. Each bath consisted of 1 L of solution that each coated two or three substrates. The solution comprised of deionized water, nickel sulphate ($NiSO_4$) as a source of nickel, and sodium hypophosphite ($NaPO_2H_2$) as a reducing agent. For composite baths, 1 g of NiTi alloy nanopowder was added per 1 L of solution. The nanopowder was characterized by 99.9% 60 nm and had a Ni:Ti ratio of 1:1. It was supplied by US Research Nanomaterials Inc. (Houston, TX, USA). Prior to plating, the substrates were cleaned by submerging for five minutes in an alkali solution with a composition of 30 g/L $Na_3PO_4$, 50 g/L $Na_2CO_3$, and 30 g/L NaOH which was heated to 85 °C. This was followed by the sample being rinsed with distilled water. Then, the substrates were submerged into room temperature 20 vol% sulfuric acid solution for 10 s for the surface activation. The substrates were then rinsed with distilled water again before being placed into a Ni-P plating solution.

Short and long deposit times were used for both the Ni-P and Ni-P-NiTi to create coatings of various thicknesses. The Ni-P-coated samples utilized a one-hour plating time and a three-hour plating time. For the Ni-P-NiTi composite coatings, before the deposit in the composite bath, they were first submerged in a Ni-P pre-coating solution for 5 min to increase the coating's adhesion. They were then immediately placed into the Ni-P-NiTi solution and remained submerged for their deposit duration.

*2.2. Coating Characterization*

Before erosion–corrosion testing, the cross-section of each coating was examined using a Hitachi S-4700 Scanning Electron Microscope (SEM) and energy dispersive spectrometry (EDS). EDS mapping verified the elemental distributions and their concentrations. After testing, both the surface and cross-sections of the coatings were examined using SEM and EDS.

Micro-hardness tests were done using an NANOVEA PB1000 mechanical tester on the surface of a monolithic Ni-P coating and a nanocomposite Ni-P-NiTi coating. A Vickers indenter was used with an applied load of 6 N. The tests were repeated multiple times over the surfaces to ensure the reproducibility of the results.

*2.3. Slurry Pot Erosion–Corrosion (SPEC) Test*

Pure corrosion, pure erosion, and erosion–corrosion tests were conducted on Ni-P and Ni-P-NiTi of various thicknesses in a slurry pot erosion–corrosion (SPEC) tester. Their results were compared to the AISI 1018 steel substrate. Figure 1 shows a schematic of an SPEC unit. A 4 L glass vessel held the samples and the slurry that was impelled by a motor-driven impeller. The impeller speed was 900 RPM and the slurry temperature was 45 °C for all the tests. It is important to note that slurry pot erosion corrosion tests were conducted at 45 °C to simulate the oil sand processing operation (i.e., hydro-transport, separation, etc.) where the temperature ranges from 40 to 50 °C. The slurry had a mixture of 3.5 wt% sodium chloride (NaCl) as a corrosive medium and 35 wt% AFS 50–70 silica sand for the erosion. Deionized water was used to prepare the test slurries. The samples were mounted in epoxy

so that only the test surface was exposed. Each coating type had three samples used for this experiment. One sample was for pure erosion ($E_0$), one for erosion–corrosion (EC), and the other one was used for pure corrosion and erosion-enhanced corrosion. A three-electrode cell with a Gamry PC4/750 potentiostat to allow for an electrochemical assessment and cathodic protection control. In this study, the linear polarization technique is used for pure corrosion and potentiodynamic polarization is used for the erosion-enhanced corrosion measurement. During the linear polarization resistance tests for pure corrosion, a potential 30 mV more negative than the corrosion potential was applied at a sweep rate of 0.6 V/h, while the erosion-enhanced corrosion potentiodynamic polarization tests were conducted at a potential 250 mV below and above the corrosion potential ($E_{corr}$). Electrochemical tests were conducted in situ vs. the saturated calomel electrode (SCE) and Pt counter electrode. Linear polarization is a non-destructive way of estimating the instantaneous corrosion rate of a metal, and its use for the pure corrosion test preserves the material's surface so that the erosion-enhanced corrosion test can be performed on the same sample. While potentiodynamic polarization is typically considered to be a destructive technique, it yields a more precise corrosion current and therefore is preferred for the erosion-enhanced corrosion test since the destructive nature is not a factor. Both linear polarization resistance and potentiodynamic polarization were performed twice and the results presented are the average corrosion rate from these two measurements.

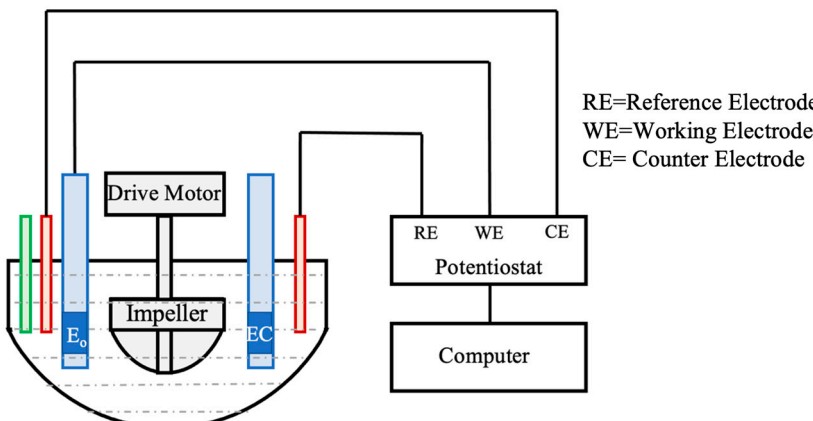

**Figure 1.** Schematic of the slurry pot erosion–corrosion arrangement.

The specimen dimensions are $18 \pm 0.5$ mm $\times$ $10 \pm 0.5$ mm $\times$ $5 \pm 0.5$ mm, where 18 mm $\times$ 10 mm is the test surface which constitutes a test area of 1.80 cm$^2$.

Natural angular silica sand was used as abrasive. Figure 2 shows the SEM micrograph and particle size distribution of the abrasive (average size 724–823 μm). The erodent particle-target impact velocities were determined by briefly exposing polished soft metal (pure copper, annealed at 450 °C) specimens in a dilute slurry of spherical particles. The normal particle impact velocity was then calculated from the depth of the resulting craters [26]. The experimentally derived normal impact velocity during the slurry pot erosion–corrosion test is 0.76–2.86 ms$^{-1}$ [27]. Within the current test setup, the typical abrasive particles impingement angles vary from 10° to 90°.

The corrosion rates of pure corrosion were conducted using the polarization resistance technique and did not degrade the surface. This allowed the same sample to then be tested for erosion-enhanced corrosion. The corrosion rates from both these tests allowed for the synergistic effects of corrosion to be evaluated, and then to find the synergistic effects of erosion. The pure corrosion used a pure 3.5 wt% sodium chloride (NaCl) solution with no silica sand, while erosion-enhanced corrosion and erosion–corrosion used the slurry.

Both the erosion–corrosion tests and pure erosion tests ran for six hours each, and the material loss rate was calculated from the test duration and the change in the sample weight before and after the test. A high precision micro-balance scale with a reading accuracy of 0.01 mg to measure the material losses of the samples before and after testing. Pure erosion

used the same slurry mixture as erosion–corrosion, but the samples were cathodically protected with 0.5 V below the open circuit potential during the test. That allowed for the material loss to be exclusively from the erosion.

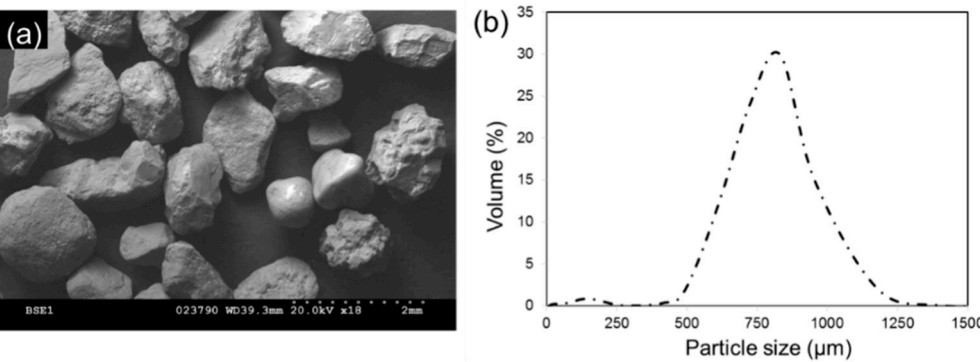

**Figure 2.** (**a**) SEM micrograph of natural angular silica sand and (**b**) particle size distribution (average size 724–823 µm).

The material loss rate was calculated in $cm^3/h/cm^2$. According to ASTM G119-09, the erosion–corrosion synergy is expressed through Equation (1), where $K_{ec}$ is the material loss under erosion–corrosion conditions, $K_{eo}$ is the material loss due to pure erosion, $K_{co}$ is the material loss due to pure corrosion, and $K_s$ is the material loss due to synergy [28].

$$K_{ec} = K_{eo} + K_{co} + K_s, \tag{1}$$

Similarly, the total material loss rate can be expressed through Equation (2), where $K_e$ is the total erosion rate, $K_c$ is the total corrosion rate, $\Delta K_e$ is the corrosion-enhanced erosion rate, and $\Delta K_c$ is the erosion-enhanced corrosion rate [28].

$$K_{ec} = K_e + K_c = K_{eo} + K_{co} + \Delta K_e + \Delta K_c, \tag{2}$$

By combining Equations (1) and (2), the synergy material loss rate and its components can be expressed as shown below [28].

$$K_s = \Delta K_e + \Delta K_c = K_{ec} - (K_{eo} + K_{co}) \tag{3}$$

$$\Delta K_c = K_c - K_{co}, \tag{4}$$

$$\Delta K_e = K_s - \Delta K_c, \tag{5}$$

The images of the coating surfaces after being subjected to rather pure erosion or erosion-enhanced corrosion were taken using a Keyence laser confocal microscope. A Ni-P coating and Ni-P-NiTi coating after erosion–corrosion were selected to have their surfaces inspected using SEM imaging and EDS mapping analysis.

## 3. Results

### 3.1. Coating Characterizaiton

The SEM micrographs of the cross-sections confirmed that the coating is well adhered to the substrate, shown in Figure 3 at 1000× magnification. In each sample, the coating adhesion is visible by a distinct change in the colouring between the substrate and the coating. Figure 2a is the thick Ni-P coating and Figure 2b is the thin Ni-P coating. The lighter colouring at the top shows the uneven surface that later was polished off to create a smooth surface for testing.

Figure 4a,b show 1000× magnification SEM images of thick and thin Ni-P-NiTi composite coatings, respectively. Each coating shows relatively uniformly distributed dark circles with a well-defined interface. Those dark circles are the NiTi nanoparticles in the Ni-P matrix. Both thick and thin coatings also show a uniform adherence to the steel substrate.

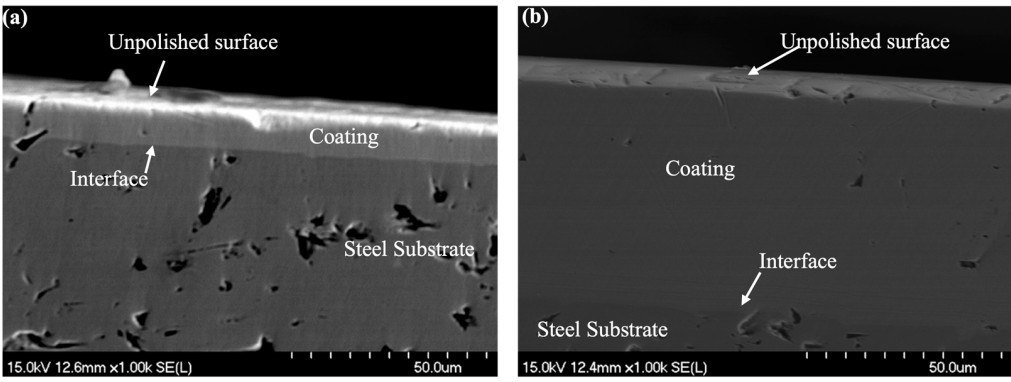

**Figure 3.** (**a**) Cross-section SEM image of thin Ni-P coating at 1000× magnification and (**b**) cross-section SEM image of thick Ni-P coating at 1000×.

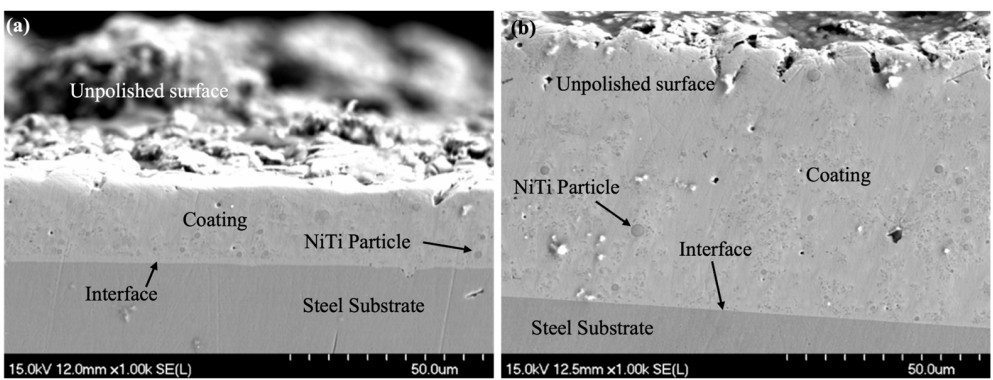

**Figure 4.** (**a**) Cross-section SEM image of thin Ni-P-NiTi composite coating at 1000× magnification and (**b**) cross-section SEM image of thick Ni-P-NiTi composite coating at 1000× magnification.

An SEM image of the as-deposited composite coating in Figure 5 shows a rough surface. It appeared to be due to the particles settling in different areas. This surface was lightly polished down to have an even surface for testing. After polishing, the sample's thicknesses were determined to be 70 μm thick Ni-P-NiTi, 60 μm thick Ni-P, 12 μm thick Ni-P, and 25 μm thick Ni-P-NiTi.

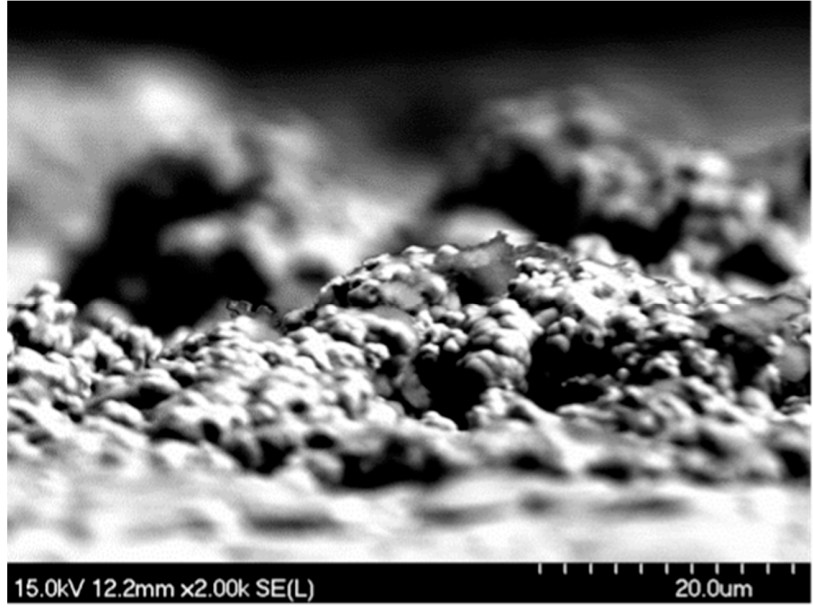

**Figure 5.** View of the rough surface SEM image of thin Ni-P-NiTi composite coating at 2000× magnification.

The compositions of the coating were found by EDS mapping on the cross-sections. The Ni-P had a defined interface between the coating and the substrate. An EDS map of the thinner Ni-P coating is shown in Figure 6. Figure 7 depicts an EDS map of the thinner Ni-P-NiTi coating, which clearly showed that the nanoparticles are embedded in the matrix. A summary of the average coating compositions is shown in Table 2.

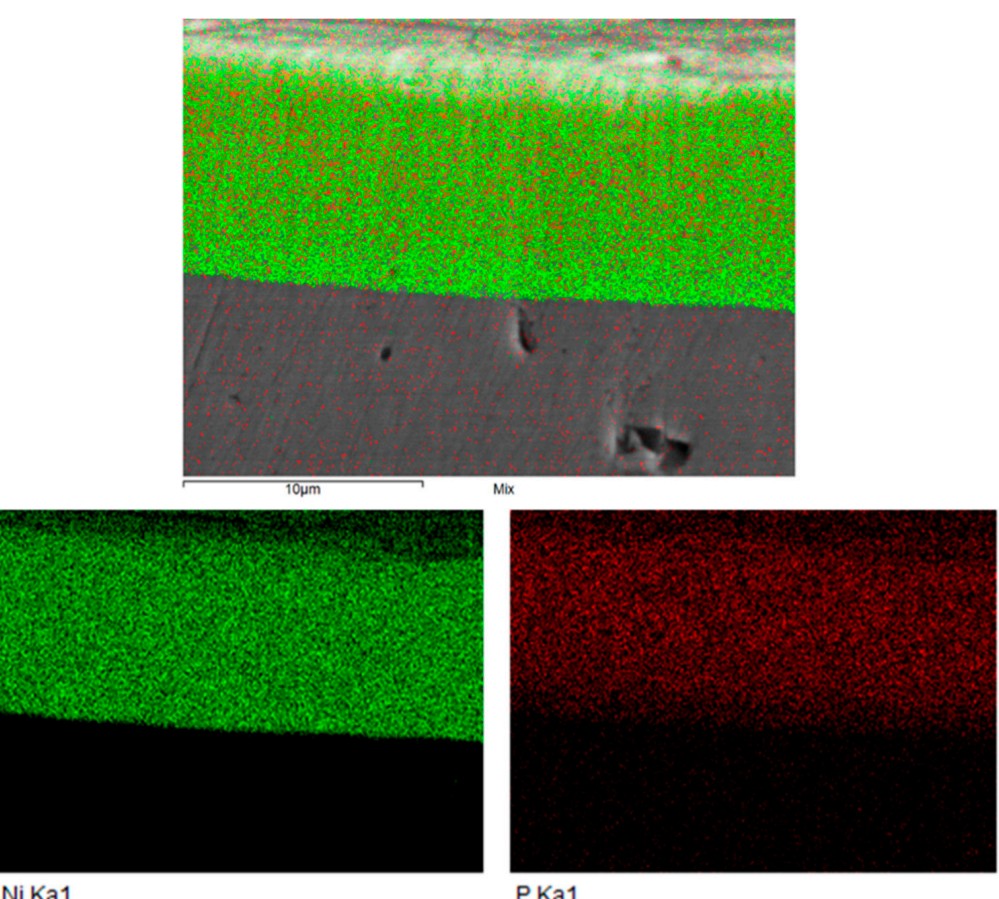

**Figure 6.** EDS map of thin Ni-P coating cross-section.

**Table 2.** Coating Compositions from EDS Analysis.

|  | Thin Ni-P | Thick Ni-P | Average Ni-P | Thin Ni-P-NiTi | Thick Ni-P-NiTi | Average Ni-P-NiTi |
|---|---|---|---|---|---|---|
| Nickel | 95.49% | 94.31% | 94.90% | 92.51% | 92.365% | 92.44% |
| Phosphorous | 4.595% | 5.69% | 5.14% | 6.125% | 6.015% | 6.07% |
| Titanium | 0% | 0% | 0% | 1.365% | 1.62% | 1.49% |

Using the average weight percent of titanium found by the EDS mapping of the coating and the Ni-Ti ratio of the powder, the weight percent of NiTi present in the composite coatings was calculated as shown in Equation (6).

$$\text{wt\%NiTi} = \%\text{Ti} + \left( {54.08}/{45.92} \right)\%\text{Ti}, \tag{6}$$

The thinner composite was found to be Ni-P-2.97wt%NiTi, while the thicker composite was Ni-P-3.53wt%NiTi. Therefore, the average composite composition would be Ni-P-3.25wt%NiTi.

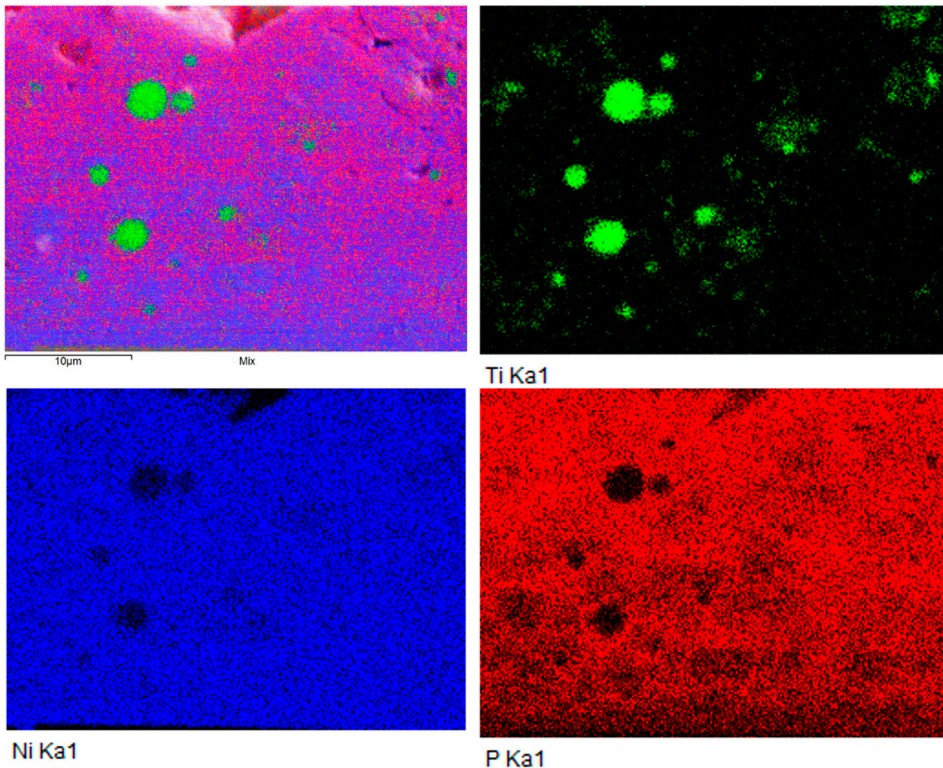

**Figure 7.** EDS map of thin Ni-P-NiTi coating cross-section.

Figure 8 shows the average Vickers micro-hardness of a Ni-P and Ni-P-NiTi coating. The error bar is the standard deviation of all the measurements taken on the sample. Ni-P had seven indentations and Ni-P-NiTi had six indentations. The typical hardness of AISI 1018 steel is 1.7–2 GPa, much lower than the average hardness that was measured on both coatings. The Ni-P coating hardness was found to be an average of $5.75 \pm 1.90$ GPa, within the typical Ni-P hardness range of 5–6.5 GPa. The Ni-P-NiTi coating had a hardness lower than the typical Ni-P range, which was $3.55 \pm 0.93$ GPa. However, it was higher than the typical NiTi alloy hardness which ranges from 2.8 to 3.2 GPa.

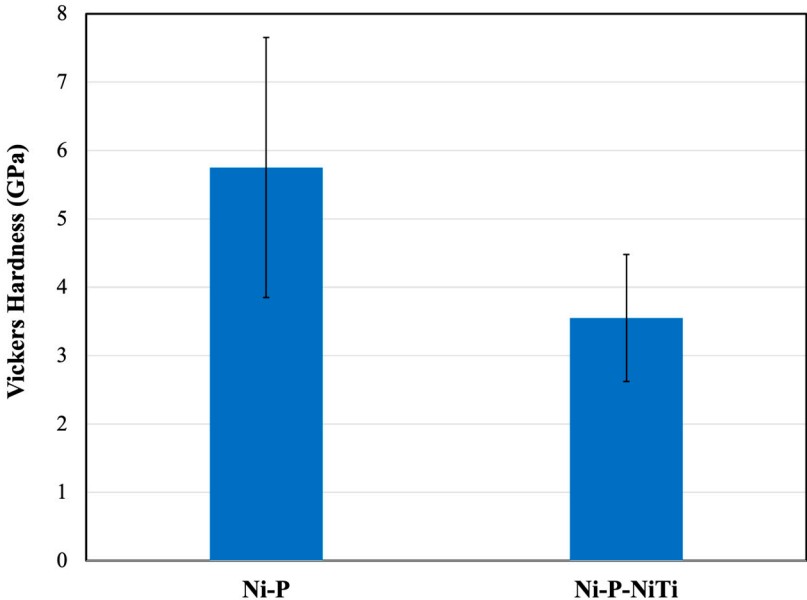

**Figure 8.** Average Vickers micro-hardness of Ni-P and Ni-P-NiTi.

### 3.2. Erosion–Corrosion Performance

Figure 9 shows the material loss rates for the AISI 1018 steel substrate, the monolithic Ni-P coatings, and the composite Ni-P-NiTi coatings. The material loss rates are for erosion–corrosion ($K_{ec}$), erosion only ($K_{eo}$), corrosion only ($K_{co}$), and total synergy ($K_s$). As hypothesized, AISI 1018 had the highest material loss rates during erosion–corrosion, pure erosion, and pure corrosion while the Ni-P coatings had the lowest material loss rates. The 12 μm thick Ni-P coating had the highest erosion–corrosion and pure erosion resistance, however, the 60 μm thick Ni-P coating had the highest corrosion resistance.

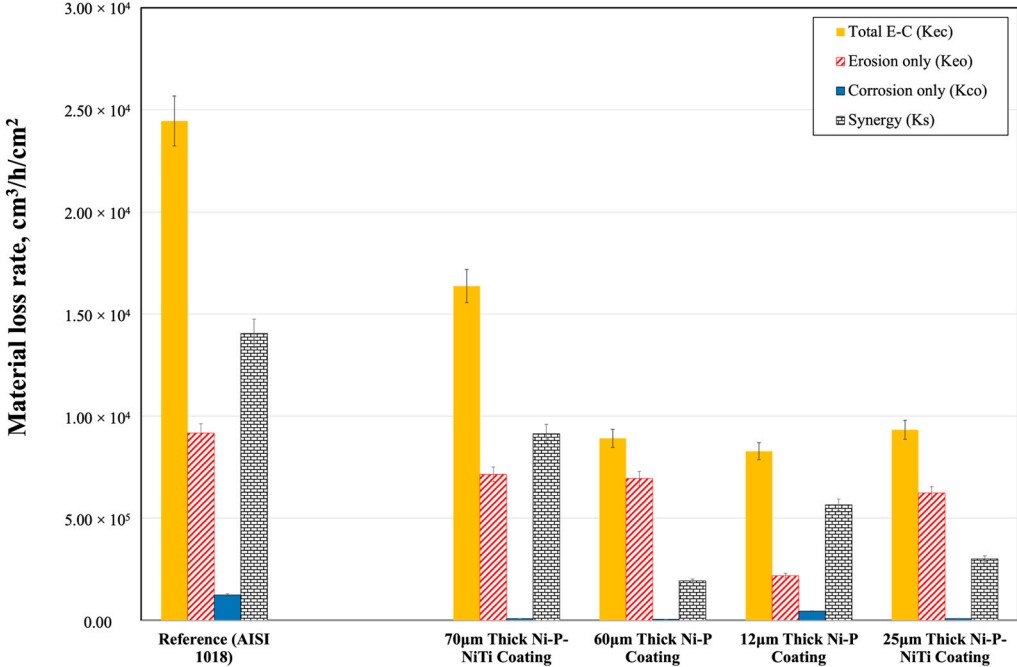

**Figure 9.** Material loss rates in erosion–corrosion experiments.

The total synergistic effect was found by simply subtracting the pure erosion and pure corrosion rates from the erosion–corrosion rate, as described in Equation (3). Both erosion–corrosion and pure erosion had their material loss rate calculated from the measured sample's mass loss and test duration. However, to be able to study the effects of corrosion, material loss rates were extrapolated from potentiodynamic polarization curves for erosion-enhanced corrosion and linear polarization for pure corrosion. Both linear and potentiodynamic polarization plots use a log scale for the current density to be able to compare the two plots.

After pure corrosion and erosion-enhanced corrosion tests, the data were plotted with respect to the potential ($E_{corr}$) vs. the current density ($i_{corr}$). Figures 10 and 11 show the plots for pure corrosion and erosion-enhanced corrosion, respectively. Figure 10 was generated from a linear polarization test and the corrosion current was determined using the Stern–Geary equation. Figure 11 was generated using potentiodynamic experiments and the corrosion current was determined from the Tafel slope. The error bar represents the standard deviation of two erosion–corrosion tests. The corrosion rate, CR, was calculated using Equation (7), where EW is the sample's equivalent weight, and D is the density of the sample. The data from the pure corrosion and erosion-enhanced corrosion are summarized in Tables 3 and 4, respectively.

$$CR = \frac{i_{corr} \times EW}{D} \times 3270, \tag{7}$$

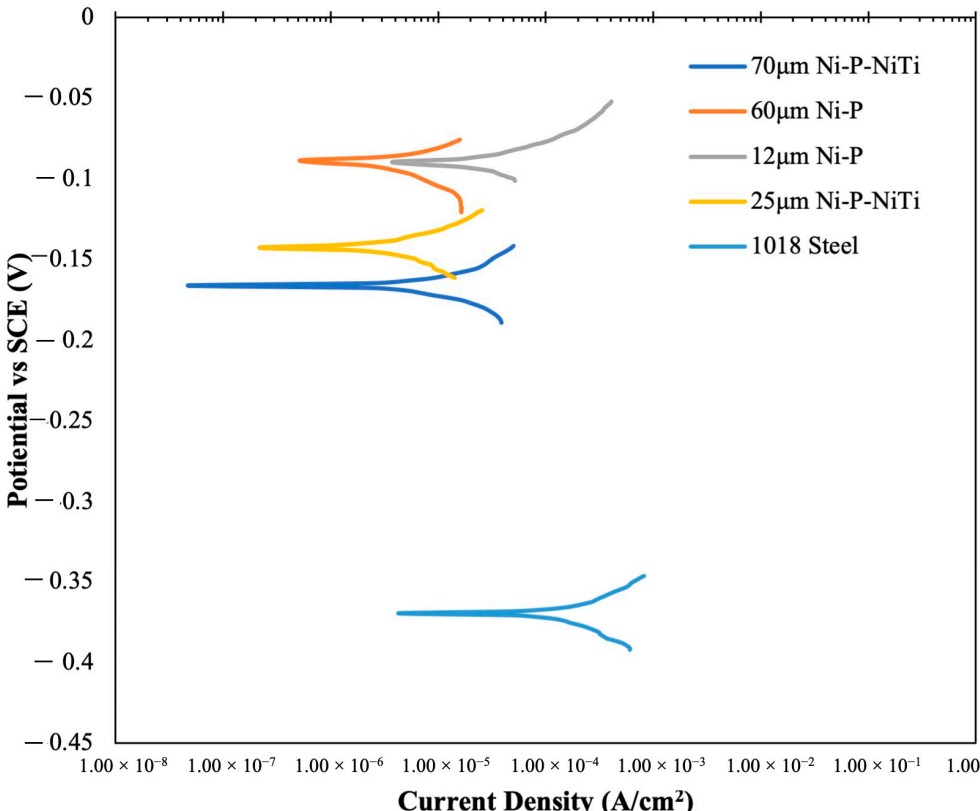

**Figure 10.** Pure corrosion linear polarization plotted as log current vs. potential.

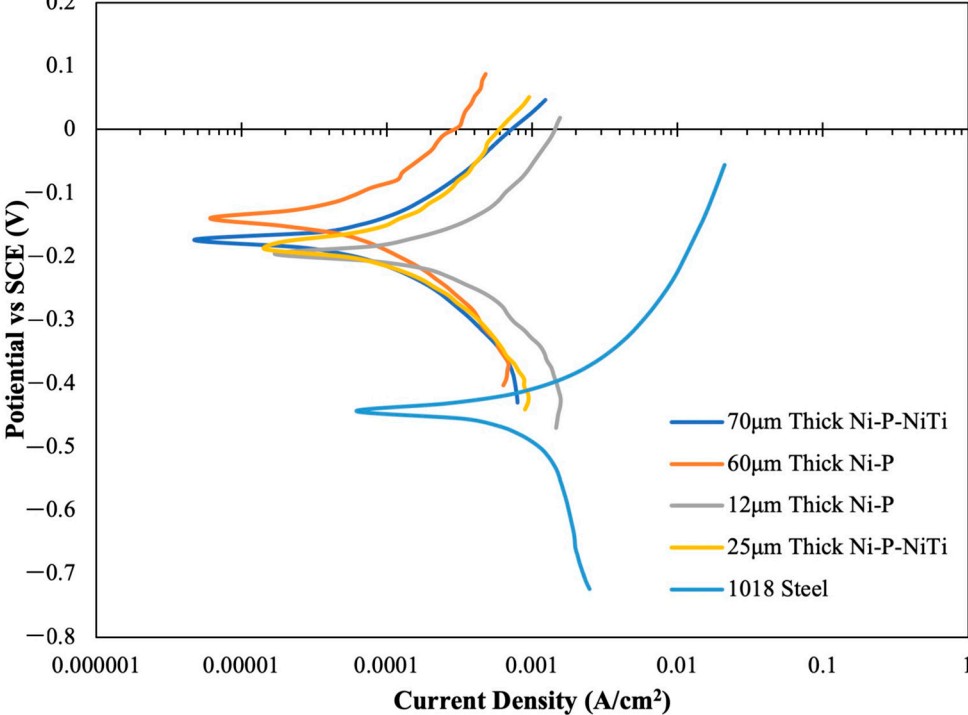

**Figure 11.** Erosion-enhanced corrosion potentiodynamic polarization plot.

**Table 3.** Pure Corrosion Results.

| Pure Corrosion | $i_{corr}$ (amp/cm$^2$) | $E_{corr}$ (volts) | Corrosion Rate (cm$^3$/(cm$^2 \times$ h)) |
|---|---|---|---|
| AISI 1018 | $9.61 \pm 0.07 \times 10^{-5}$ | $-3.70 \pm 0.18 \times 10^{-1}$ | $1.24 \pm 0.009 \times 10^{-5}$ |
| 70 µm thick Ni-P-NiTi | $6.73 \pm 0.59 \times 10^{-6}$ | $-1.66 \pm 0.0009 \times 10^{-1}$ | $7.87 \pm 0.69 \times 10^{-7}$ |
| 60 µm thick Ni-P | $4.37 \pm 1.46 \times 10^{-6}$ | $-9.63 \pm 1.08 \times 10^{-2}$ | $4.50 \pm 0.85 \times 10^{-7}$ |
| 12 µm thick Ni-P | $3.78 \pm 0.55 \times 10^{-6}$ | $-1.14 \pm 0.35 \times 10^{-1}$ | $4.45 \pm 0.65 \times 10^{-6}$ |
| 25 µm thick Ni-P-NiTi | $7.18 \pm 2.45 \times 10^{-6}$ | $-1.44 \pm 0.003 \times 10^{-1}$ | $8.37 \pm 2.86 \times 10^{-7}$ |

**Table 4.** Erosion-Enhanced Corrosion Results.

| Erosion-Enhanced Corrosion | $i_{corr}$ (amp/cm$^2$) | $E_{corr}$ (volts) | Corrosion Rate (cm$^3$/(cm$^2 \times$ h)) |
|---|---|---|---|
| AISI 1018 | $7.88 \pm 0.26 \times 10^{-4}$ | $-4.44 \pm 0.02 \times 10^{-1}$ | $1.02 \pm 0.03 \times 10^{-4}$ |
| 70 µm thick Ni-P-NiTi | $1.00 \pm 0.12 \times 10^{-4}$ | $-1.76 \pm 0.004 \times 10^{-1}$ | $1.17 \pm 0.14 \times 10^{-5}$ |
| 60 µm thick Ni-P | $6.27 \pm 1.99 \times 10^{-5}$ | $-1.47 \pm 0.10 \times 10^{-1}$ | $7.31 \pm 2.32 \times 10^{-6}$ |
| 12 µm thick Ni-P | $3.95 \pm 2.82 \times 10^{-4}$ | $-1.96 \pm 0.03 \times 10^{-1}$ | $4.64 \pm 3.31 \times 10^{-5}$ |
| 25 µm thick Ni-P-NiTi | $2.17 \pm 0.52 \times 10^{-4}$ | $-1.85 \pm 0.05 \times 10^{-1}$ | $2.54 \pm 0.61 \times 10^{-5}$ |

The 60 µm thick Ni-P coating's corrosion rate was almost two orders of magnitude lower than the 1018 substate. The significant corrosion resistance of the monolithic coating is mostly due to its microstructure and chemical composition. Generally, amorphous structures typically have a high corrosion resistance due to their lack of grain boundaries. This is because grain boundaries are high-energy sites that are susceptible to corrosion [29–31]. Furthermore, nickel and phosphorus reaction mechanisms both contribute to preventing rapid corrosion. Initially, the phosphorus can react with water to form hypophosphite anions. It has been theorized that a layer of these anions acts as a barrier on the surface, which shields the material from the environment [32]. This inhibits the nickel's hydration reaction, which is what is needed for the active dissolution of nickel [31,32]. When nickel does react, it forms a passive layer of nickel oxide (NiO) on the surface which protects the material below from further corrosion [31–33].

During erosion–corrosion, the formation of the passive layer is greatly affected by the surface roughness introduced by erosion. The impact energy of the particles is absorbed, and the surface is deformed plastically. Plastic deformation from erosion is evident on the coating's surface. Figures 12 and 13 show the micrographs and 3D imaging of the 70 µm thick Ni-P-NiTi surface after pure erosion. The micrographs show the uneven surface, and the material displacement is quantified by the 3D images. Figure 12 shows the formation of a crater, where the particle impact presses the coating material outward to create a cavity and displaces it to form a hill. The rough edges of the crater are noteworthy, as it shows how the material was deformed. Figure 13 shows a similar crater. However, the hill has been broken off due to repeated particle impact.

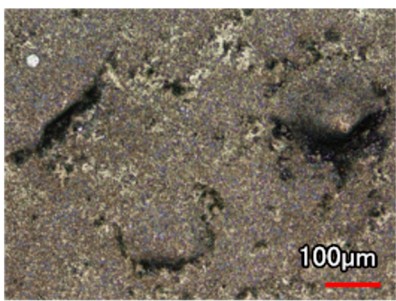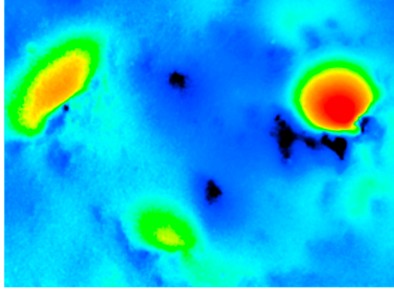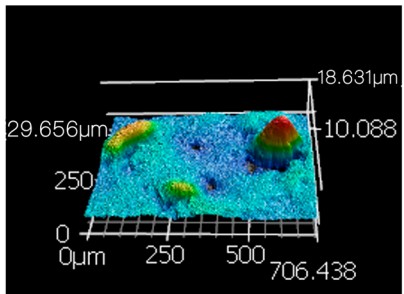

**Figure 12.** Crater formation on a pure erosion surface of 70 µm thick Ni-P-NiTi.

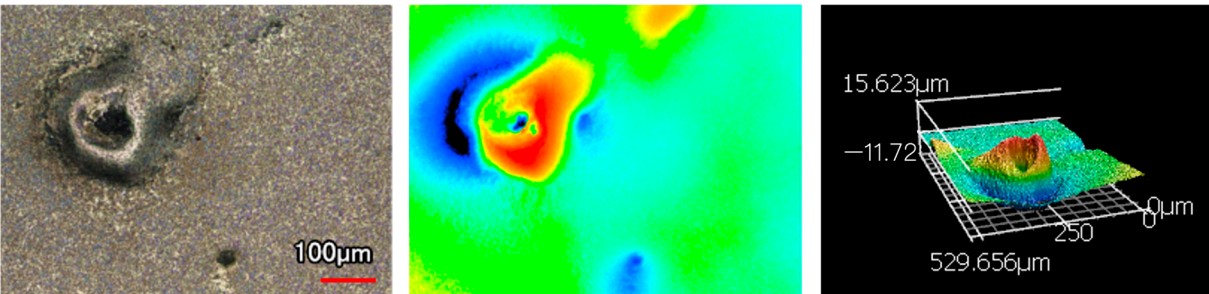

**Figure 13.** Crater with edges on a pure erosion surface of 70 μm thick Ni-P-NiTi.

The plastically deformed crater sites are visible on the erosion–corrosion surface, and the effect of corrosion is observable. The plastically deformed hill is dissolved from corrosion, along with the rough edges on the crater sites. This appears as a much smoother indent site than is seen in pure corrosion. Figure 14 shows an example on the surface of the 70 μm thick Ni-P-NiTi coating.

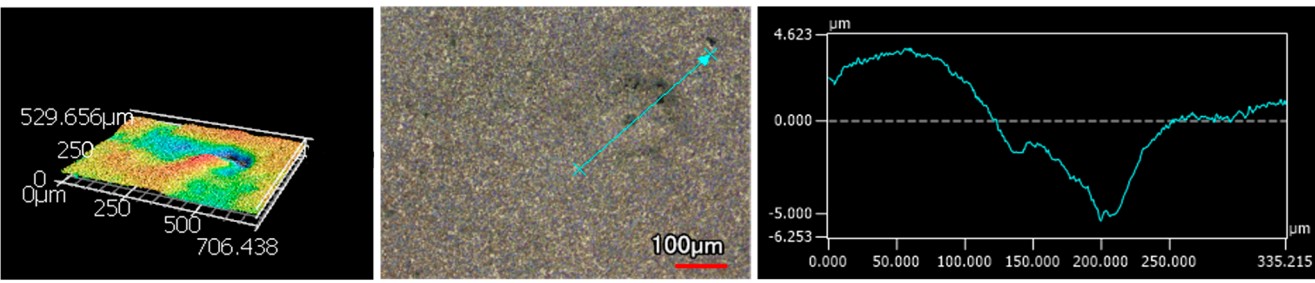

**Figure 14.** Representation of E crater from erosion influenced by corrosion on the surface of 70 μm thick Ni-P-NiTi after erosion–corrosion.

The erosion rates after 6 h of the test inside the slurry pot are shown in Figure 15. The highest erosion rate was observed for the thickest coating (i.e., 70 μm thick Ni-P-NiTi), while the thinnest coating (i.e., 12 μm thick Ni-P) had the lowest rate. When comparing all four coatings regardless of the differences in the compositions, the erosion rate increases with an increased thickness.

The erosion rate and coating thickness correlation can be explained by the residual stresses. The deformation produces compressive residual stresses and, consequently, tensile stresses at the subsurface. This cold working process modifies the mechanical properties of the material. The principle is applied in the industry as a mechanical treatment method to improve the wear resistance, which is known as shot peening [34,35].

Typically, residual stress induced during plating in the coating is tensile near the substrate and transitions into compressive stress as the thickness increases [22,23]. The initial residual stresses in the material have an effect on the consequential stresses induced by shot peening [36,37]. During erosion, which is effectively equivalent to shot peening, this thick coating surface undergoes a further compression [38,39]. However, in shot peening, the tensile stresses develop at the subsurface to accommodate the additional surface compressive stresses. The changes in the internal stresses is depicted as a schematic in Figure 16 [38]. Figure 16a, before the erosion image, shows an approximation of the distribution of the internal stresses of the as-plated thick coating. After the coating surface is subjected to compression from erosion, the new approximate distribution of the stresses is shown in Figure 16b, after the erosion image. The internal stresses are relieved from the delamination shown in Figure 16c, relieving the internal stresses. Delamination is caused by erosion in relation of the internal stresses of a thick coating. This is further depicted by a schematic in Figure 17. Conversely, thin coatings are subjected to surface compressive stresses during erosion which cancels some or all of the tensile stress developed during plating. Therefore, the thin coating has less tensile residual stress than it had as-plated, but

the thick coatings have a higher near-surface tensile residual stress. This effect is evident in the lower erosion resistance and observable wear mechanisms for thick coatings.

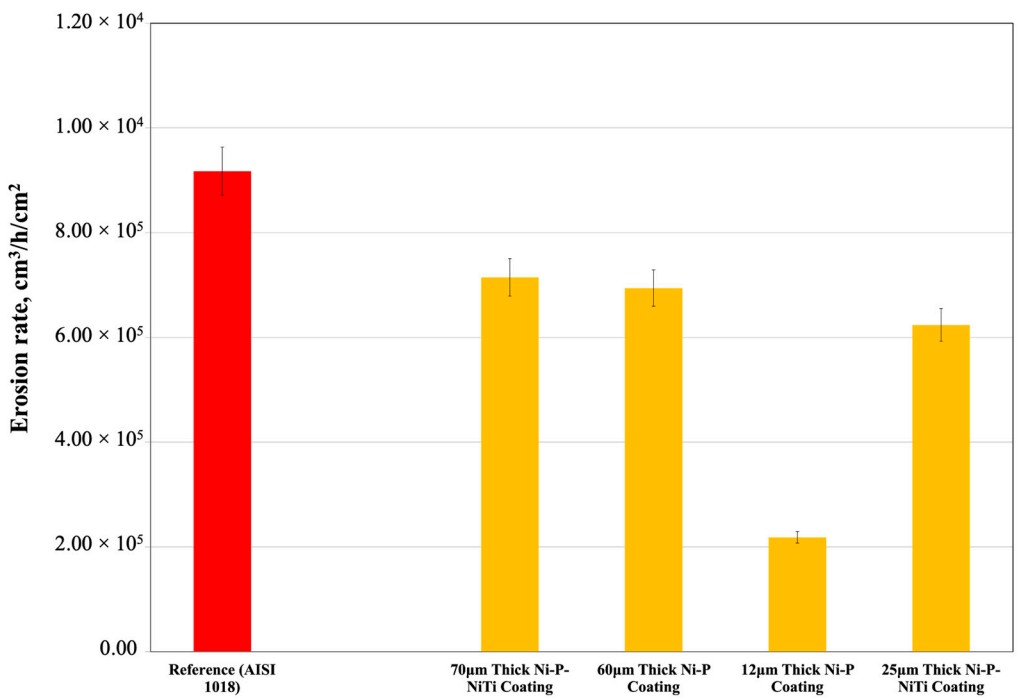

**Figure 15.** Pure erosion rates after 6 h of test inside the slurry pot.

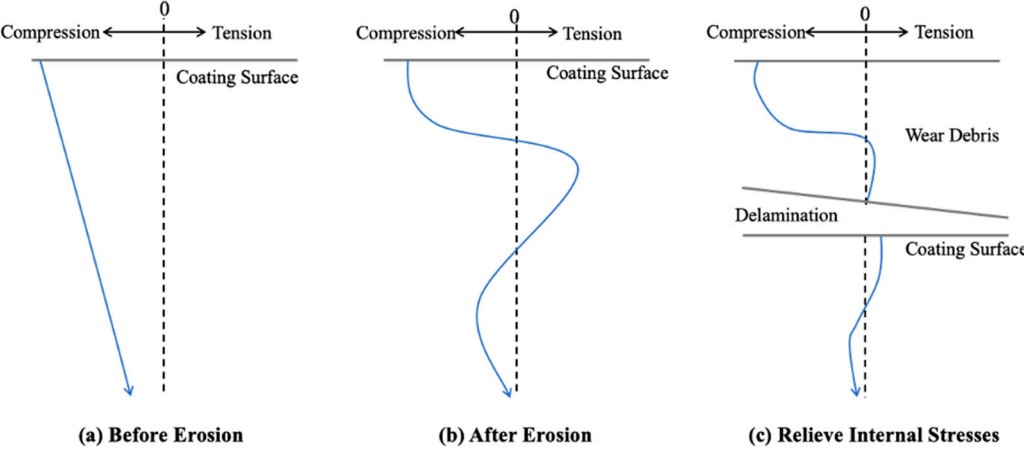

**Figure 16.** Delamination in relation to the residual stresses induced by erosion where (**a**) thick coating as plated; (**b**) thick coating after erosion; and (**c**) delamination.

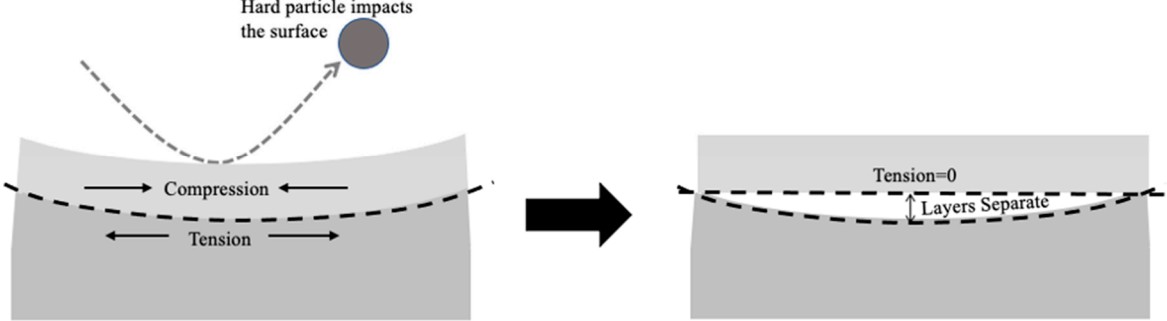

**Figure 17.** Schematic of delamination in relation to the residual stresses induced by erosion.

There was evidence of delamination in the thicker coatings. Figure 18a,b show the SEM images of the 60 μm thick Ni-P and 70 μm thick Ni-P-NiTi coatings after erosion–corrosion, respectively. Figure 18a shows the initiation of delamination in the 60 μm thick Ni-P coating. The coating is lifted from the substrate, but not fully peeled off yet. Figure 18b shows the completion of delamination on the 70 μm thick Ni-P-NiTi coating.

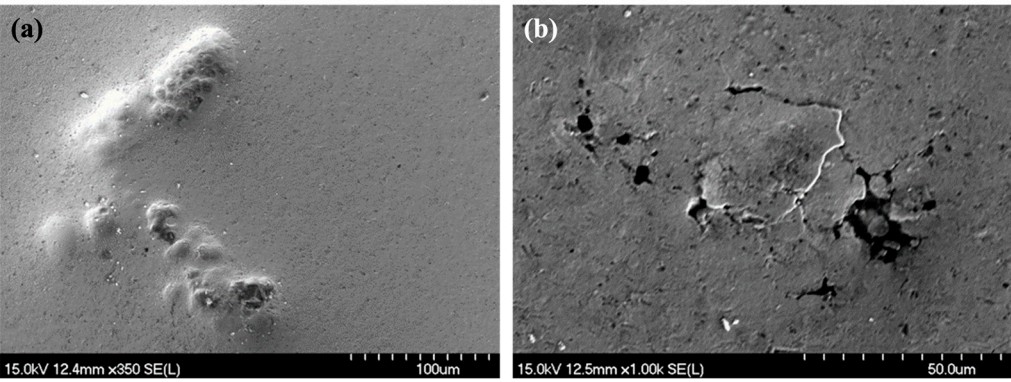

**Figure 18.** SEM image of (**a**) initial stages of delamination on the 60 μm thick Ni-P surface after erosion–corrosion and (**b**) delamination on the 70 μm thick Ni-P-NiTi surface after erosion–corrosion.

Without the shot-peening effect from erosion, pure corrosion has a different correlation between the coating thickness and the material loss rate. Figure 19 shows the corrosion-only material loss rates; the thinnest coating, 12 μm thick Ni-P, had the highest rate compared to the other thicknesses.

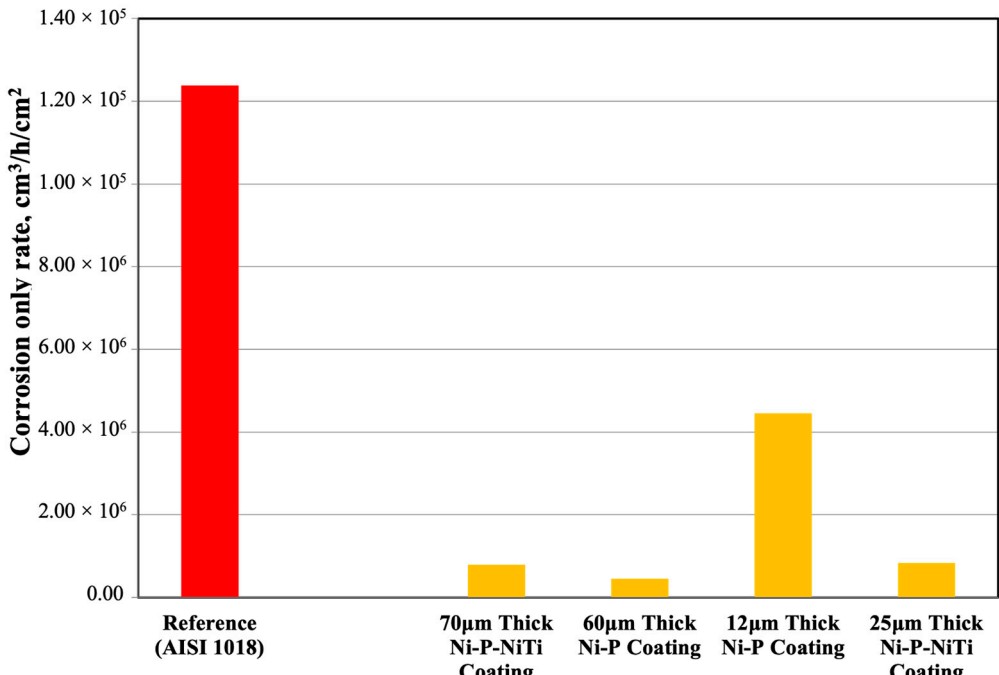

**Figure 19.** Pure corrosion rates of the coatings.

Since the thinner Ni-P coating is only 12 μm thick, it is likely that the coating fractured and exposed the steel substrate to the corrosive environment for part of the test duration. The corrosion of the steel substrate accounts for the spike in the material loss compared to the other coatings. The 60 μm thick Ni-P coating had the lowest material loss rate due to the NiO passive layer formation, its mostly amorphous microstructure, and the high thickness that prevents fractures from reaching the steel substrate. However, despite the 70 μm thick Ni-P-NiTi composite coating being slightly thicker than the 60 μm thick Ni-P monolithic

coating, it was not as effective due to the presence of the NiTi nanoparticles. Titanium has a higher reactivity to form oxide than nickel does, and therefore a higher tendency for the formation of a $TiO_2$ passive layer [14,21,40]. However, since the NiTi content is limited and dispersed throughout the matrix, the $TiO_2$ layer is thin and sporadically distributed. This means that the $TiO_2$ is not as effective at protecting the surface. The nickel is susceptible to reacting with the environment, but the amount of NiO formation is limited due to the tendency for the oxygen to combine with the titanium [14]. Consequently, the dissolution of nickel increases and allows for the 70 μm thick Ni-P-NiTi coating to have a slightly higher corrosion rate compared to the 60 μm thick Ni-P. Evidence of the sporadically distributed oxide is seen in localized corrosion sites, known as corrosion pitting. This can be seen in Figure 20, an SEM image of the 70 μm thick Ni-P-NiTi coating surface after erosion–corrosion. Notably, plastic deformation and delamination from erosion are also present.

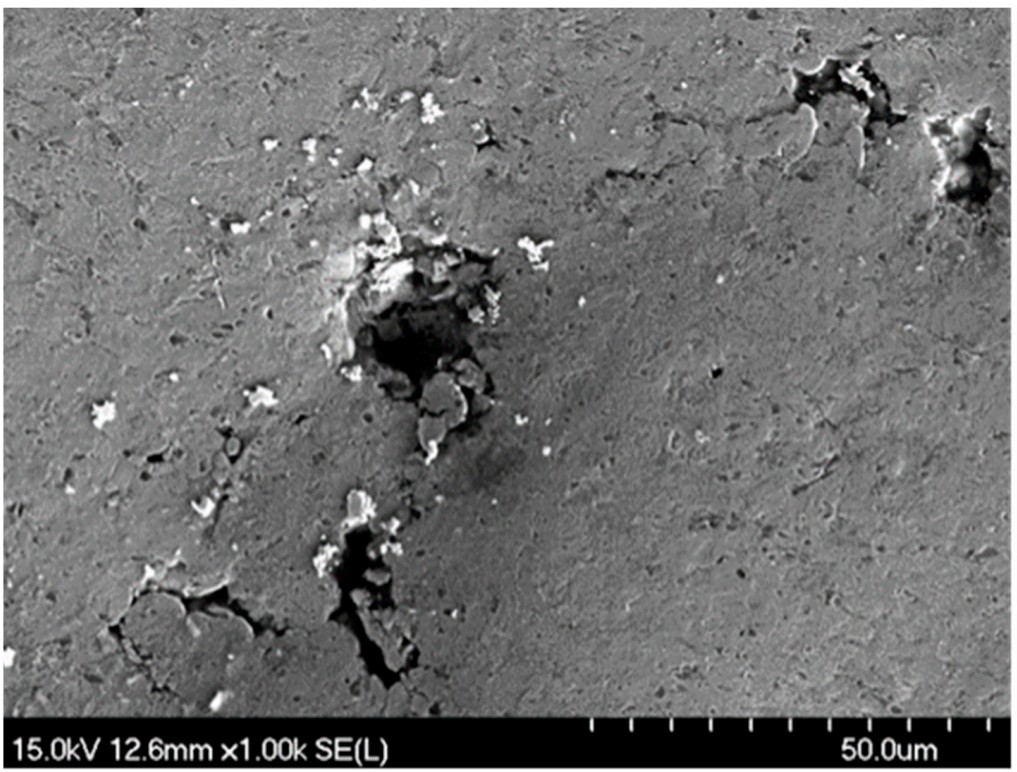

**Figure 20.** SEM image of localized corrosion on the 70 μm thick Ni-P-NiTi surface after erosion–corrosion.

EDS mapping was done after erosion–corrosion testing to confirm the presence of oxygen. Figure 21 is the EDS map of the 60 μm thick Ni-P and Figure 22 is the EDS map of the 70 μm thick Ni-P-NiTi coating.

Figure 23 is to characterize the total synergy effect on the material loss rate by its synergistic components: increase in corrosion due to erosion ($\Delta K_c$) and increase in erosion due to corrosion ($\Delta K_e$). The 70 μm thick Ni-P-NiTi had the highest synergistic effect, mostly from its significantly higher $\Delta K_e$ compared to the other coatings.

The 70 μm thick Ni-P-NiTi coating had almost 148% times the $\Delta K_e$ than the 60 μm thick Ni-P coating. Since they have close thicknesses, this infers that there is a significant effect of the presence of NiTi particles within the Ni-P matrix. As the corrosion dissolves the cold-worked surface formed from erosion, the softer coating sublayer is exposed [41]. This softer surface would be more susceptible to erosion damage, giving rise to the high erosion rate. As the surface erodes, the NiTi particles protrude from the surface [42]. The presence of these particles resulted in a higher surface roughness, while the Ni-P coating would remain smoother [43,44]. Increased surface roughness is a known effect of increased

erosion [45], which would explain why the smoother 60 μm thick Ni-P coating would not experience as drastic of an increase in $\Delta K_e$ as the 70 μm thick Ni-P-NiTi coating.

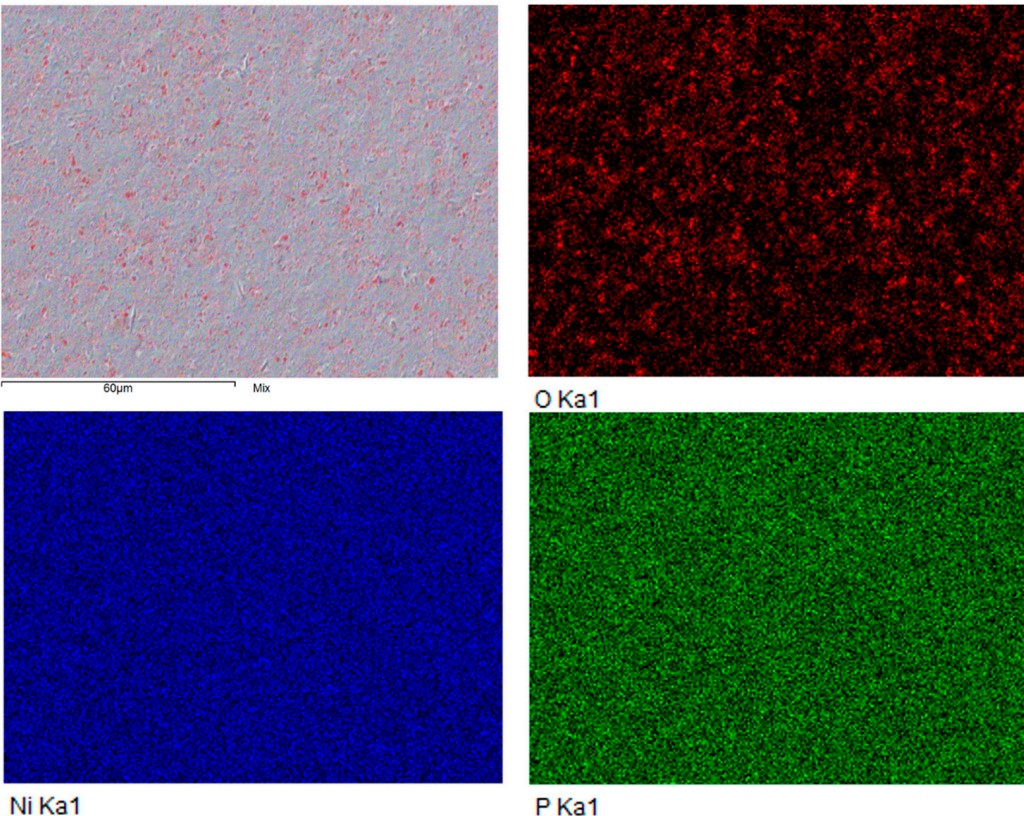

**Figure 21.** EDS mapping of 60 μm thick Ni-P surface after erosion–corrosion.

The 12 μm thick Ni-P coating had the highest $\Delta K_c$ out of all the coatings. The impacted surface from erosion particles had high stress and strain, which are known to be more anodic and, therefore, more susceptible to corrosion. The thin coating likely fractured and exposed the steel to the environment. The steel and remaining coating would continue to be impacted by erosion, therefore continuously exhibiting anodic behaviour after material loss and thus accelerating corrosion [46–48]. The surface damage and deep pits on the surface of a 12 μm thick Ni-P coating after erosion–corrosion are shown in Figures 24 and 25.

As shown previously in Figures 24 and 25, the 12 μm thick Ni-P coating showed deep pits throughout the surface after erosion–corrosion. However, this is also seen in the 60 μm thick Ni-P coating surface after erosion–corrosion. Figure 26 shows several deep pits, which are not seen in the composite coatings to the same degree. Figure 27 shows a deep crack, which was an exclusive feature unique to the Ni-P monolithic coatings.

When comparing the Ni-P monolithic coatings to the Ni-P-NiTi composite coatings, the monolithic had lower material loss rates from erosion–corrosion. However, this does not necessarily reflect on if they provide a better protection to the substrate. Deep narrow pits could show a minimal material loss, but they are more detrimental than the equivalent of evenly distributed loss. Furthermore, deep cracks would not be accounted for but would also expose the steel substrate to the environment.

Figure 28 is a representation of the overall damage that was seen in the 70 μm thick Ni-P-NiTi coating. There was no evidence of pitting corrosion and no cracks. The absence of cracking can be explained by the NiTi toughening mechanisms. This coating had the mildest damage defects from erosion–corrosion.

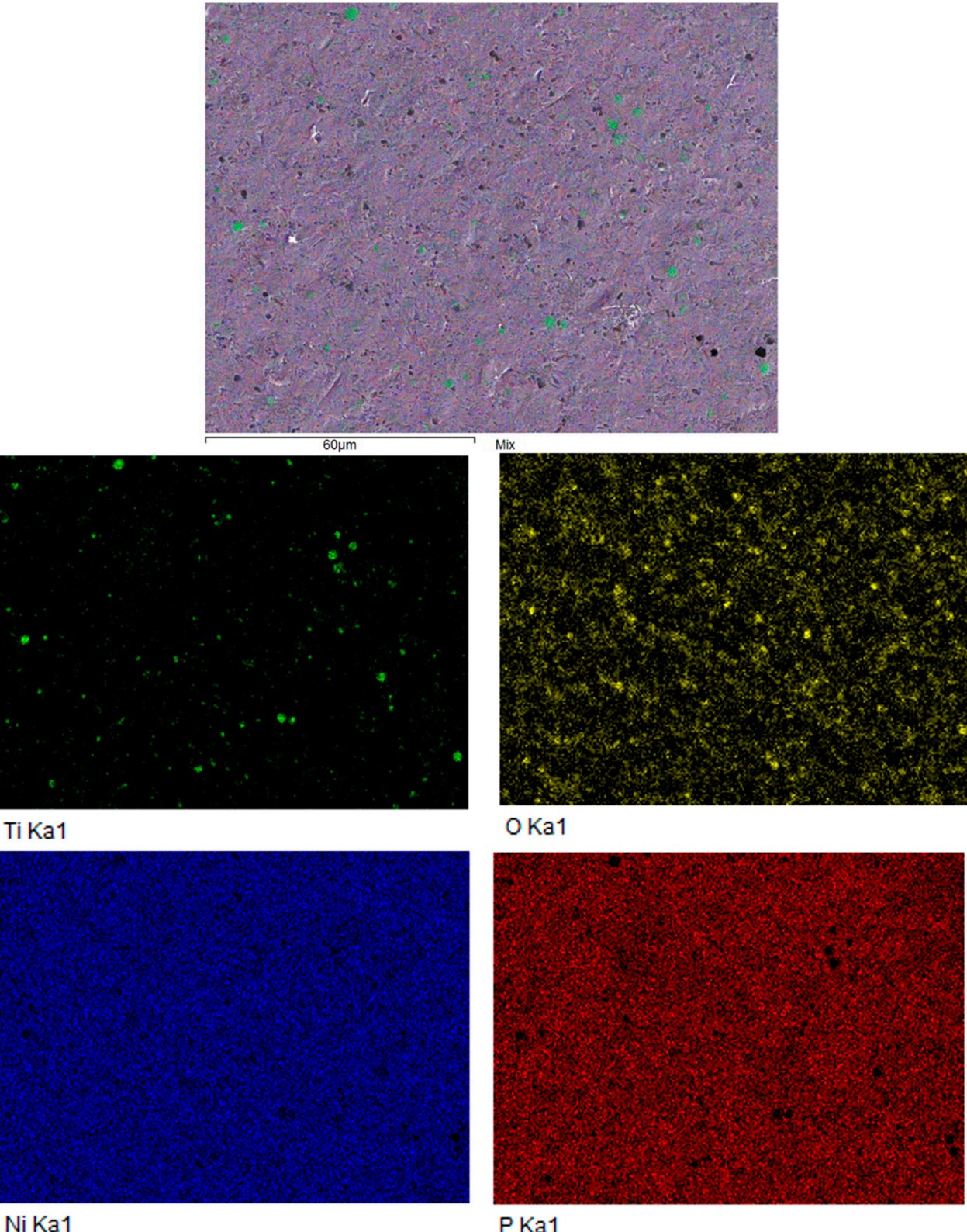

**Figure 22.** Representation EDS mapping of 70 μm thick Ni-P-NiTi surface after erosion–corrosion.

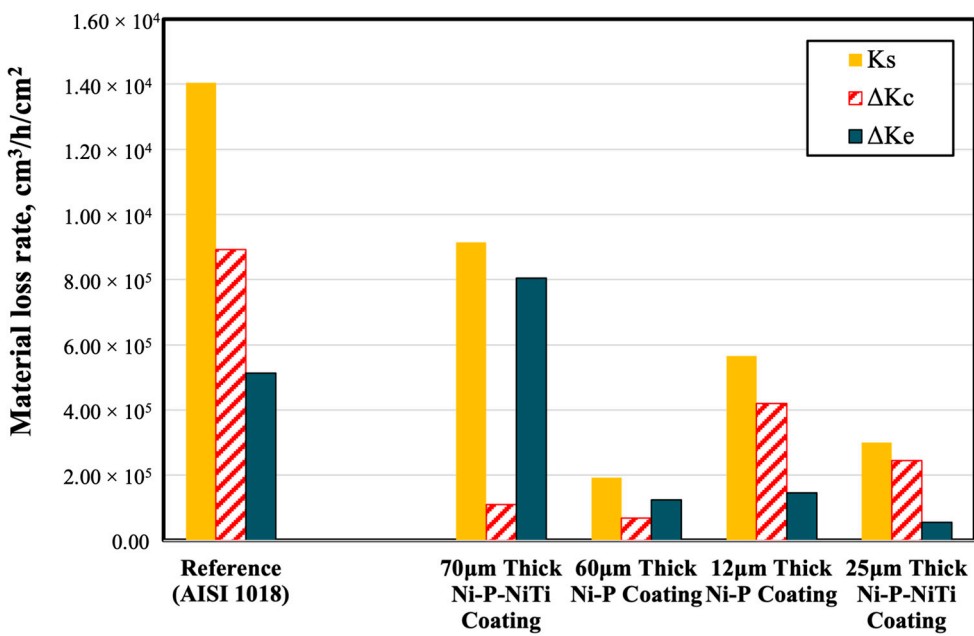

**Figure 23.** Representation synergistic material loss rates.

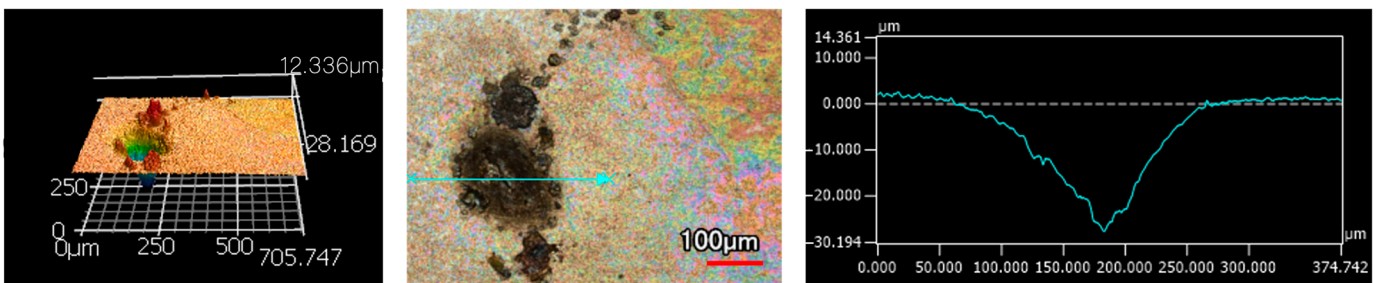

**Figure 24.** Representation deep pit from erosion–corrosion on the 12 μm thick Ni-P coating.

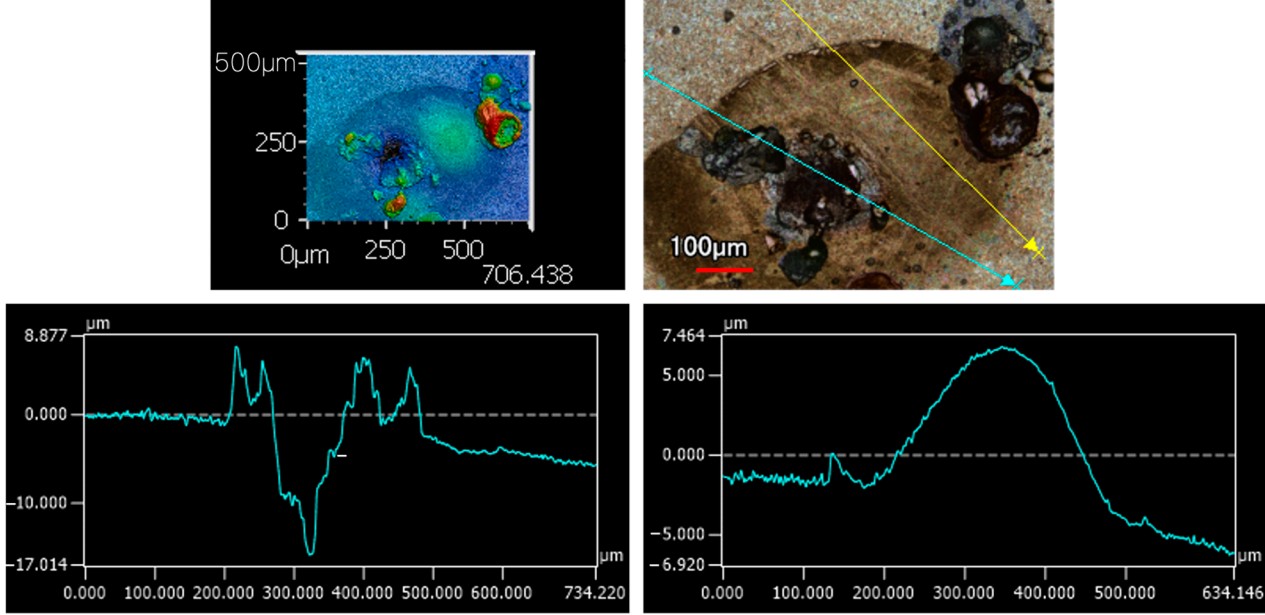

**Figure 25.** Deep pits and surface roughness from erosion–corrosion on the 12 μm thick Ni-P coating.

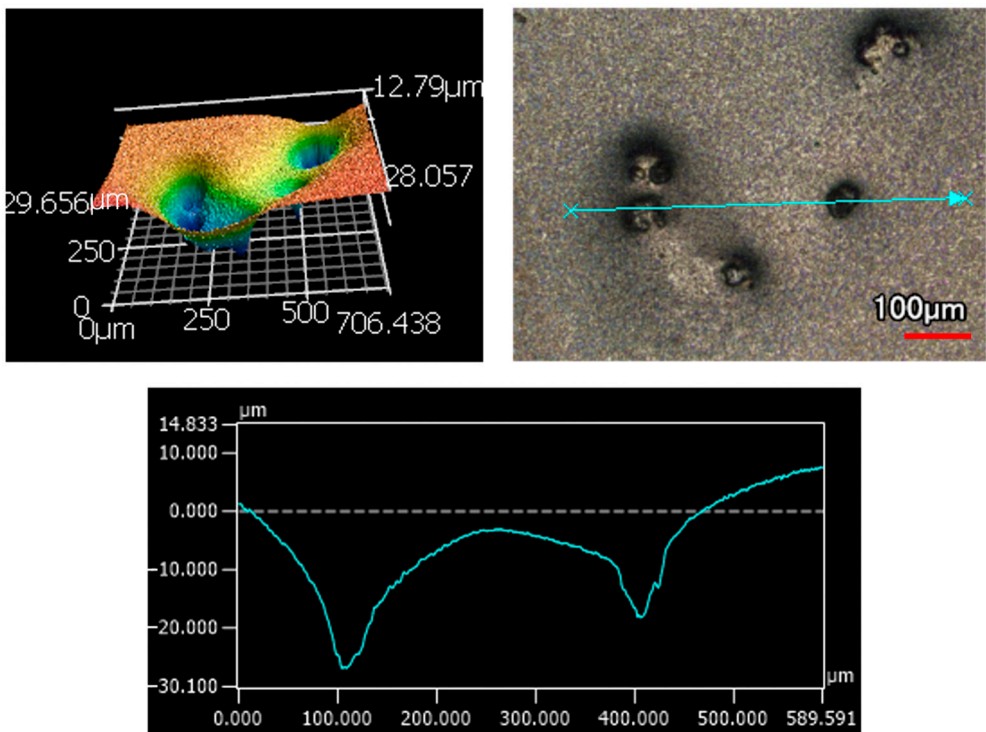

**Figure 26.** Cluster of deep pits from erosion–corrosion on the 60 μm thick Ni-P surface.

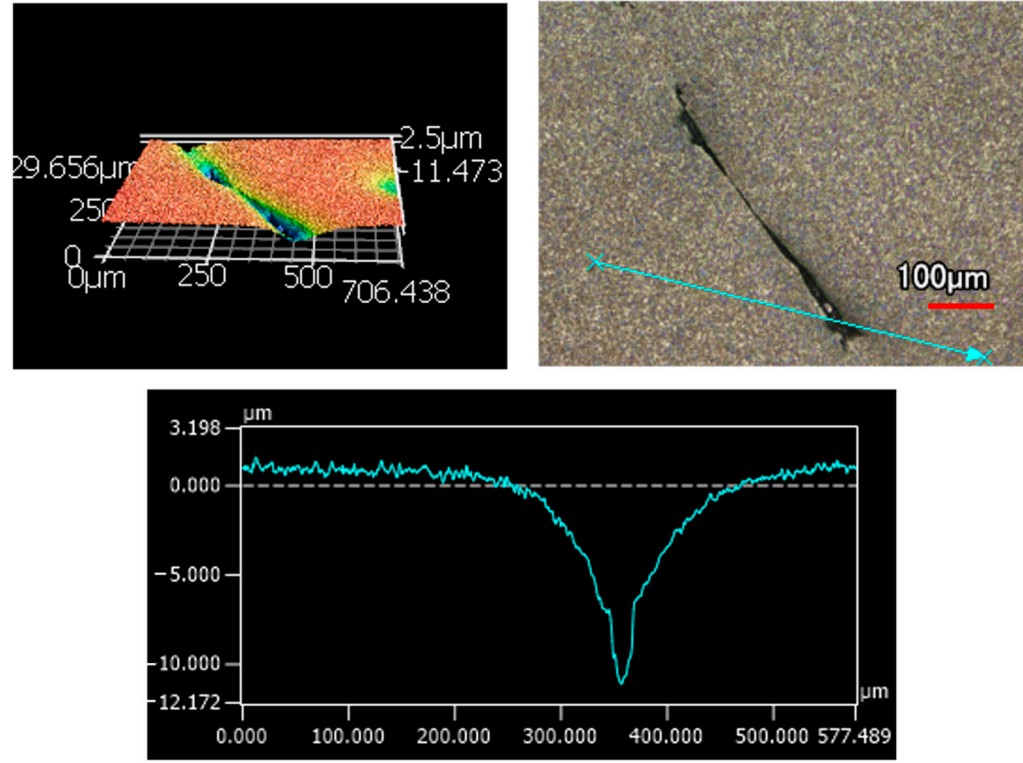

**Figure 27.** Crack from erosion–corrosion on the 60 μm thick Ni-P surface.

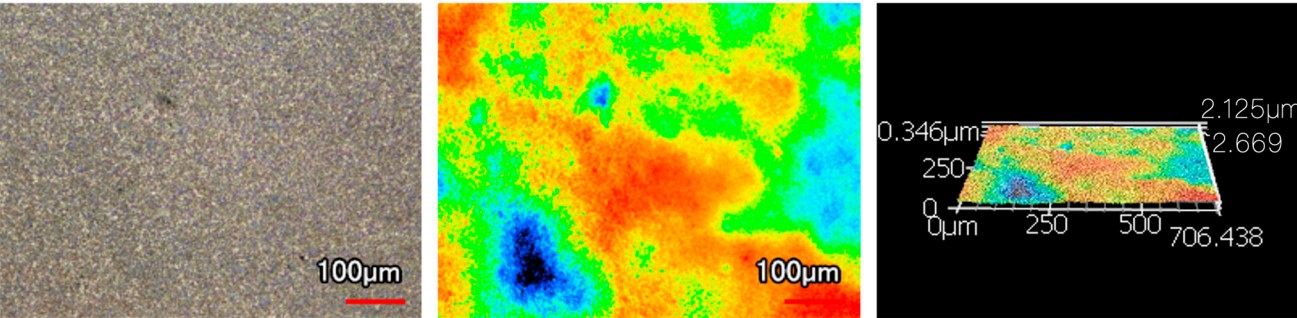

**Figure 28.** Surface of the 70 μm thick Ni-P-NiTi after erosion–corrosion.

Given the observable damage mechanisms, the Ni-P-NiTi coatings would provide better protection for the steel substrate from erosion–corrosion. However, the 70 μm thick Ni-P-NiTi coating did have the highest material loss rates, which should not be disregarded. Therefore, when accounting for both the visible damage and material loss rate, the 25 μm thick Ni-P-NiTi coating seems to have the highest erosion–corrosion resistance. Furthermore, thinner coatings can have significant cost savings in terms of time and materials. This strengthens the appeal of using the thinner Ni-P-NiTi composite coating for protecting steel pipelines from erosion–corrosion.

## 4. Conclusions

In summary, the monolithic Ni-P and Ni-P-NiTi composite coatings of various thicknesses were prepared on AISI 1018 steel substrates and were then tested for erosion–corrosion resistance. The following conclusions were made by comparing the material loss rates and observed degradation mechanisms:

- The thickness and presence of nanoparticles proved to be significant factors in the coating's performance, which is suggestive of the substantial role that the degree of residual stress has on the coating.
- The thicker coatings had more degradation from the particle impact than their counterpart thinner coating. Additionally, producing a thinner coating is more cost efficient than a thick coating due to time and materials savings.
- The monolithic Ni-P coatings had a lower material weight loss but had degradation features that are more detrimental in comparison to the higher amount of uniform material loss that was observed on the Ni-P-NiTi coatings.

Overall, of all the coatings tested, a thin Ni-P-NiTi composite coating of approximately 25 μm thickness is determined to provide the best option for the protection against erosion–corrosion for low-carbon steel pipelines.

**Author Contributions:** Conceptualization, Z.F.; methodology, R.J., Z.F. and M.A.I.; formal analysis, R.J. and M.A.I.; investigation, R.J.; resources, Z.F., M.A.I. and G.J.; data curation, M.A.I.; writing—original draft preparation, R.J.; writing—review and editing, Z.F. and M.A.I.; supervision, Z.F.; All authors have read and agreed to the published version of the manuscript.

**Funding:** This research was funded by the Natural Sciences and Engineering Research Council of Canada (NSERC), grant number RGPIN 05125-17.

**Data Availability Statement:** Data is contained within the article.

**Conflicts of Interest:** The authors declare no conflict of interest.

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
