# Peer review of "Erosion–Corrosion of Novel Electroless Ni-P-NiTi Composite Coating"

_cmd, doi:10.3390/cmd4010008_

Round 1
Reviewer 1 Report
This is very interesting, and visually attractive manuscript related with the different coatings (monolithic Ni-P coatings, and the composite Ni-P-NiTi coatings) on AISI 1018 steel.
Authors have been used lots of different techniques in characterization of the developed coating. Also, the manuscript is written in the simple and understandable manner. SEM/EDS images and analysis in the manuscript looks very attractive, clear, and sharp.
The manuscript has no significant weaknesses.
However, I have a few suggestions for improvements of its high quality even more.
In Materials and Methods section, I suggest a few additions to the text because I found that some details are missing in this version:
· please state the manufacturer of the AISI 1018 steel coupons and their chemical composition
· please write details related with the experimental conditions, such as temperature, mixing of the solution, etc.
· please add the information about SEM/EDS instrument (manufacturer and model)
· please state the manufacturer and the model of micro-hardness tester used.
· please supplement the text with data on the conditions of conducting the polarization resistance experiments in which potential range was the experiment conducted, the value of scan rate (potential change velocity in mV/s) and which potentiostat was used.
In 3. Results section:
· please delete 3.1.1. Subsubsection title
· on figure 9 and 10 please add the curve for the referent AISI 1018 steel sample (without coating)
· Figure 10 shows potentiodynamic polarization curves which were conducted in the higher potential area (from the graph in the potential area of ±250 mV around Ecorr). This method is not mentioned in the Materials and Methods section, as should be.
· Please add the value of corrosion potential for the AISI 1018 in the Table 1 and Table 2.

Reviewer 2 Report
Generally is OK, but in the text there were found some mistakes that concern language and describes. The list below shows only a few the most important points and questions, that concern unknowns or mistakes:
1. What is the size of added nanoparticles?
2. Is it possible to present the results of the average composite compositions in the table?
3. In line 122 there is a mistake with adnotation "Error! Reference source not found."
4. How many times the test of microhardness were conducted? In the text (line 118) there is only the following information: "The tests were repeated multiple times over the surfaces to ensure reproducibility of results." It is important also to check the standard deviation.
5. In the point 2.1. there is written, that "Rectangular AISI 1018 steel coupons were used as substrate." and the results also contain this material - it is OK. But why in conclusions there is information that "In summary, monolithic Ni-P coatings and Ni-P-NiTi coatings were prepared on API X100 and AISI 1018 steel substrates."?
6. In line 70 there is the following mistake: effect4ed. Please correct this word.
7. Is it possible to show in the table the components concentrations of substance for deposition of coating?
8. Why exactly the coatings thickness are 12, 25, 60 and 70um? It would be better, if for the thick layers Ni-P-NiTi and Ni-P the thickness would be 60 or 70um, and for the thin layers the thickness would be 12 or 25um. What is the main reason?
9. Is it possible to write a bit more conclusions, for example the results can be described in relation to lifespan of low-carbon steel petroleum pipelines? Because the article in abstract and introduction begins with description about pipelines.
10. Please, make sure and check carefully again if the language and style everywhere is correct.
Reviewer 3 Report
The paper presents some interesting results on electroless coatings containing nanoparticles. Whilst there is value in studying this, there are concerns about the methodology implemented. There does not appear to be any justification for the conditions chosen and these conditions do not appear to represent the application represented by the study (more clarity in the introduction may help improve this).
Corrosion data appears to have been obtained too simplistically to make the conclusions that have been obtained. Measurements have not been reported over time and have implemented techniques that have limitations.
There are no repeats of any erosion-corrosion experiments. Erosion-corrosion experiments typically have a significant amount of uncertainty due to the complexity of experiments. A lack of repeats means the data cannot be fully trusted.
There are also too many figures – 27 is excessive and a number of these could be removed. It is also not clear what the benefits of adding nanoparticles are - erosion-corrosion rates are higher in some cases when they are present. The most critical factor appears to be coating thickness, not the addition of nanoparticles.
In addition there are a number of points that need to be addressed:
Line 32 – corrosive species should be written in full before using chemical symbols, e.g. chloride ions
Line 70 – typo in ‘effected’
Line 74 – First initial not required for Jensen et al. (and all other citations throughout)
Method
Line 99 – subscripts required for chemical symbols
Line 101 – heated alkali solution requires more information. What was this solution and what temperature was it heated to?
Line 118 – how many times is multiple times? A minimum number could be stated, e.g. a minimum of 10 indentations.
Line 121 – results were compared to bare steel – how was the substrate prepared? A brief note on the method should be added. Was the steel prepared for experiments using the same grinding papers for example
Line 122 – formatting error
Line 123 – ‘heled’ typo
Line 125 – why was 45°C chosen? This would be considered a fairly low temperature for many of the applications this study is representing.
Line 126 – define NaCl (as in line 32)
Section 2.3 – more detail is required for the experimental setup. Was deionised water used?
What velocity of particle impact does 900 rpm result in? And what are the typical particle impact angles on the test specimens?
What size and general shape were the sand particles? SEM images of the particles are useful if they were obtained. The average sand particle size in microns should be included in the description.
Were experiments completed in aerated environments? Oil and gas would typically consist of a CO2 or H2S environment as acknowledged on line 32.
Fig 1 – it is not clear what Eo and EC mean.
Polarization resistance has limitations for application with coatings. Were EIS measurements considered? This technique would be much more suitable for a coating application.
Can the authors be certain that cathodic protection did not fundamentally change the properties of the surface? Or lead to any embrittlement? Why were experiments not performed under a nitrogen or argon-saturated environment to assess erosion without the need for cathodic protection?
Results
Figure 2 – the coating labelled as ‘thin’ appears to be thicker. These images are on the same scale.
Figure 8 – y axis scale labels should be changed. No decimal places are shown. Why are there are no error bars on the plots? Erosion and erosion-corrosion experiments are typically complicated due to the vast number of parameters that influence results. Repeat measurements are essential.
Fig 9 – why is linear polarisation data shown on a log scale? Are corrosion rates determined from one measurement? The corrosion rate may vary significantly over time.
Fig 10 – why has this been measured over a different potential range? And when was this measurement performed in the experiment? It is a destructive technique. As the erosion rates are high compared to corrosion, it may be possible that erosion-enhanced corrosion changes considerably over time.
Fig 14 – no repeats to give confidence in the conclusion that erosion rate increases with thickness. How much of this degradation is degradation of the substrate and how much is degradation of the coating?
Fig 15 and Fig 16 are not necessary and can be described in discussion
There is a lot of speculation around degradation mechanisms, but repeat data has not been obtained and therefore there is no confidence in those conclusions.
Statements such as line 371-373 are very speculative and not backed up by physical evidence in the experiments or literature.
Conclusions
These need to be rewritten. It is not clear what the main conclusions from this work. Was the addition of nanoparticles beneficial? It does not appear to be in much of the erosion-corrosion data.
Reviewer 4 Report
The authors studied the erosion-corrosion performance of Ni-P-Ni Ti Composites coatings using slurry pot. The newly developed coatings with solid particles could provide protection of pipeline steels when transporting slurry. Generally, the paper is well written and organized. The manuscript could be published after minor revision.
(1) The manuscript contains 27 figures which is too much. Some figures could be move to a supplemental material to be more readable.
(2) Scale bars of many figures are missing, please carefully check these figures.
(3) The formation of the erosion-corrosion pits is not well clarified. The following articles could be references
Flow accelerated corrosion and eroison-corrosion behavior of marine carbon steel in natural seawater, npj Materials Degradation, (2021)5:56
Probing the initiation and propagation processes of flow accelerated corrosion and erosion corrosion under simulated turbulent flow conditions, Corrosion Science, 2019.05, 151, 163-174
Exploring the effects of sand impacts and anodic dissolution on localized erosion-corrosion in sand entraining electrolyte. Wear. 478-479, 203907 (2021).
Experimental study on erosion-corrosion of carbon steel in flowing NaCl solution of different pH, Journal of Materials Research & Technology, 2022. 20, 4432-4451
(4) The future application of the new coatings should be introduced.
(5) The conclusions should be divided into several items.
Round 2
Reviewer 1 Report
Dear authors,
You answered on all my suggestions and made appropriate corrections, so I don't have any remarks on this manuscript.
I must congratulate you on this excellent manuscript.
I look forward to seeing your next investigations.
Author Response
Thank you for your kind words and your approval of the manuscript is much appreciated.
Reviewer 3 Report
Thank you for updating the manuscript. However, there are a few responses that require clarification.
Response 14 – what do error bars represent? Were repeat tests performed? If so, how many tests and how were error bars determined?
Response 16 – this response does not entirely address the comment. Two different methods have been used to obtain corrosion rates in Fig 10 and Fig 11. Why? This is irrelevant of any standard and is related to consistency in the methodology. The scan ranges have not been specified in the Methodology as being any different for determining pure corrosion and erosion-enhanced corrosion.
Reviewer 4 Report
The revised version could be accepeted.
Author Response
Thank you for your approval of the manuscript.
Round 3
Reviewer 3 Report
Thank you for the revisions. However, my comment regarding linear polarization and potentiodynamic polarization has still not been addressed and appears to be misunderstood. Both of these techniques are able to determine pure corrosion and erosion-enhanced corrosion in exactly the same way, so why is there a difference in the methodology? Destructive/non-destructive criteria or variation/lack of variation with time etc. applies to both corrosion and erosion-enhanced corrosion measurement in this context.
Furthermore, the methods to determine corrosion rates are different for each technique, but it appears the same methodology has been applied. The use of potentiodynamic polarization to determine icorr using Tafel extrapolation is significantly more accurate and cannot be found in the same way for linear polarization. Linear polarization would traditionally be used to determine a polarization resistance (explaining why I said this should be plotted on a linear scale, rather than a log scale), with corrosion currents then determined using the Stern-Geary equation in combination with Tafel constants. The lower polarization range prevents the direct determination of icorr accurately. Therefore, is it critical that this difference is explained because it may have a notable influence on the results.
Also, upon reviewing this data again I have noticed that the units for current density in Fig 10 and Fig 11 should be in Amps/cm^2 not Amps/cm.
The comment that erosion-enhanced corrosion does not change over time has not been proven in this manuscript and was a comment I previously made. Erosion-enhanced corrosion can vary over time but I accept that this was not the aim of this manuscript and can be addressed in future studies
Author Response
Thank you for the time you have took to ensure the paper is up to standard, it is greatly appreciated that you are being thorough with the communication to ensure an adequate manuscript.
Please see the attachment that has responses to your concerns along with revisions that hopefully address them appropriately.

Round 4
Reviewer 3 Report
Thank you for the updates. My final recommendation is that I appreciate the reasoning for choosing these methods but the application of LPR is still not correct to determine corrosion rate. The implementation of Tafel extrapolation to determine corrosion currents is incorrectly applied. This is acceptable for potentiodynamic polarization but not for LPR due to the smaller scan range.
Corrosion current can be determined by applying the Stern Geary equation using polarization resistance determined from LPR measurements. This method requires the determination of Tafel constants from Tafel extrapolation - these could potentially be determined from other experiments (e.g. Fig 11 or repeats of Fig. 10 with a higher scan range), assumed (120 mV/dec is typical but might be inaccurate for this application) or found from literature. Justification can be provided for the values chosen. Further experiments are not required unless this is preferred to determine Tafel constants.
Author Response
Understandable, however Figure 10 was generated from a linear polarization test and the corrosion current was determined using Stern-Geary equation then the data was plotted with a log scale for comparability to Figure 11. Figure 11 was generated using potentiodynamic experiments and the corrosion current was determined from Tafel slope.
The following sentence has been added on lines 308-311 of the manuscript for clarity:
"Figure 10 was generated from a linear polarization test and the corrosion current was determined using Stern-Geary equation. Figure 11 was generated using potentiodynamic experiments and the corrosion current was determined from Tafel slope."